# Anabolic Peptide-Enriched Stealth Nanoliposomes for Effective Anti-Osteoporotic Therapy

**DOI:** 10.3390/pharmaceutics14112417

**Published:** 2022-11-09

**Authors:** Sagar Salave, Dhwani Rana, Hemant Kumar, Nagavendra Kommineni, Derajram Benival

**Affiliations:** 1National Institute of Pharmaceutical Education and Research (NIPER), Ahmedabad 382355, India; 2Center for Biomedical Research, Population Council, New York, NY 10065, USA

**Keywords:** PTH (1-34), osteoporosis, nanoliposomes, QbD, Box Behnken design, MG-63 cells

## Abstract

The objective of the present work was to develop PTH (1-34)-loaded stealth nanoliposomes (PTH-LPs) by employing the use of the Quality by Design (QbD) approach. Risk identification was carried out using the Ishikawa fishbone diagram. PTH-LPs were optimized using Box Behnken Design, a type of response surface methodology to examine the effect of independent variables on dependent variables such as particle size and percentage entrapment efficiency (%EE). Design space was generated for PTH-LPs to reduce interbatch variability during the formulation development process. Furthermore, a cytotoxicity assay, cell proliferation assay, calcium calorimetric assay, mineralized nodule formation, and cellular uptake assay were carried out on MG-63 osteoblast-like cells. The results obtained from these procedures demonstrated that lipid concentration had a significant positive impact on particle size and %EE, whereas cholesterol concentration showed a reduction in %EE. The particle size and %EE of optimized formulation were found to be 147.76 ± 2.14 nm and 69.18 ± 3.62%, respectively. Optimized PTH-LPs showed the sustained release profile of the drug. In vitro cell evaluation studies showed PTH-LPs have good biocompatibility with MG-63 cells. The cell proliferation study revealed that PTH-LPs induced osteoblast differentiation which improved the formation of mineralized nodules in MG-63 cells. The outcome of the present study conclusively demonstrated the potential of the QbD concept to build quality in PTH-LPs with improved osteoanabolic therapy in osteoporosis.

## 1. Introduction

Osteoporosis is characterized by a progressive decline in bone mineral density [1,2]. It progresses silently, and patients come to know about the disease generally when a painful and severe fracture occurs. The efficacy of several pharmacotherapeutics has been investigated for the management of this skeletal disorder. PTH (1-34) (Teriparatide) is an FDA-approved peptide-based drug that stimulates bone formation via stimulating osteoblasts [3,4]. This anabolic drug is mainly administered to patients suffering from postmenopausal and drug-induced osteoporosis. The recommended therapeutic dose for the drug is 20 µg injected subcutaneously once a day for two years [3,5,6]. However, its short half-life severely restricts the treatment efficiency. In humans, the reported half-life of the drug is 5 min intravenously [7]. 

Therefore, various drug delivery strategies have been explored to modulate PTH (1-34) release [8,9,10]. For instance, Altaani et al. have developed a nanoemulsion of PTH (1-34) using a polyelectrolyte complexation approach [11]. PTH (1-34)-loaded PLGA microspheres were developed by Eswaramoorthy et al. and explored to suppress osteoarthritis progression in rats [12]. Baskaran et al. used PLGA microspheres containing PTH (1-34) for sustained release of the drug in a rat model. Another study by Rajalakshmanan et al. demonstrated the release of PTH (1-34) from PLGA microspheres in rats for 3 weeks [13]. Apart from microspheres, nanoformulations have also been explored for PTH (1-34) delivery. Dave et al. have developed hydroxyapatite-based nanorods of PTH (1-34) to enhance the anabolic effects [14], and Jaji et al. have explored aragonite calcium carbonate nanocrystals for potential PTH (1-34) delivery [15]. Narayanan et al. have explored chitosan nanoparticles [16], thiolated chitosan nanoparticles [17], and pegylated nanoparticles for the delivery of PTH (1-34) in osteoporosis [18]. Kristensen et al. have investigated the effect of the coadministration of insulin and PTH (1-34) at different pH by employing a cell-penetrating peptide [19]. To the best of our knowledge, pegylated nanoliposomes have not been explored as a drug carrier system for PTH (1-34).

Liposomes remain as one of the most preferred drug-delivery systems [20,21,22,23]. Different properties of liposomes like biocompatibility, low toxicity, biodegradability, and aptitude to entrap both hydrophilic and hydrophobic drugs make liposomes widely accepted nanocarriers in drug delivery [24,25]. In addition, the ease of manufacturing at a large scale compared to other nano-based drug delivery systems makes liposomes interesting not only for investigational purposes but also as a commercially viable drug-delivery system.

Several polymers are being investigated to improve the systemic circulation time of liposomes [26]; however, poly-(ethylene glycol) (PEG) has been widely explored among polymeric materials as a steric stabilizer. It is a linear polyether diol with many useful characteristics, such as solubility in aqueous and organic media, biocompatibility [27], good excretion kinetics [28], and very low immunogenicity [29]. These versatile and important characteristics allow its use in a variety of applications in drug delivery, as well as in the biomedical field. 

The development of a pharmaceutical dosage form is based on several critical aspects, including the selection of formulation variables and process parameters that can affect the quality of the final dosage form. A more systematic approach for developing a pharmaceutical formulation is Quality by Design (QbD), which is a risk- and knowledge-based quality management approach. It is “a systematic approach to development that begins with a predefined objective and emphasizes product and process understanding and process control, based on sound science and quality risk management” [30]. Regulatory agencies encourage pharmaceutical industries to adopt a QbD approach to build quality into the product through design efforts, from product conceptualization to commercialization [31,32]. 

In our previous study, we screened material attributes and process parameters affecting PTH (1-34)-encapsulated lipid vesicles using Taguchi standard orthogonal array L8 design [33]. The present work discusses the development of PTH (1-34)-loaded stealth nanoliposomes (PTH-LPs) using the QbD concept to understand the formulation and process variables for improving the quality of nanoformulation in terms of particle size and % entrapment efficiency (%EE), which are critical parameters that affect the performance of PTH-LPs. Further, various cellular assays were performed to check the effectiveness of optimized PTH-LPs in osteoporotic treatment. 

## 2. Materials and Methods

PTH (1-34), paraformaldehyde, alizarin red solution, DAPI (4′,6-Diamidino-2-phenylindole dihydrochloride), and cholesterol was procured from Sigma-Aldrich (Bangalore, India). Hydrogenated phosphatidylcholine (HSPC) and N-(carbonyl methoxy polyethyleneglycol-2000)-1,2-distearoylsn-glycero-3-phosphoethanolamine (Na-salt; MPEG-2000-DSPE) were obtained as gift samples from Lipoid GmbH (Ludwigshafen am Rhein, Germany). Absolute ethanol was procured from Shree Chalthan Vibhag Khand, Uddyog Sahakari Mandli Ltd., Surat, India, and Sodium chloride (NaCl) from HiMedia Laboratories Pvt. Ltd. (Mumbai, India). Potassium chloride (KCl), sodium hydrogen phosphate (Na_2_HPO4), potassium dihydrogen phosphate (KH_2_PO4), acetonitrile, formic acid, and isopropyl alcohol were procured from Fischer Scientific (Mumbai, India). NBD-PE (N-(7-nitrobenz-2-oxa-1,3-diazol-4-yl)-1,2-dihexadecanoyl-sn-glycero-3-phosphoethanolamine, triethylammonium Salt), trypsin, and fetal bovine serum (FBS) were purchased from Invitrogen^TM^, Thermo-Fischer Scientific (Mumbai, India). Ultra-pure water from Millipore Milli-Q (Synergy UV) water purification system (Merck Millipore) was used throughout the study. All other reagents used were of analytical grade and were used without further processing. MG-63 osteoblast-like cells were procured from the National Center for Cell Science (NCCS), Pune, India. 

### 2.1. QbD Approach in Formulation Development

The entire QbD-based study involves the following elements: (a) Define the Quality Targeted Product Profile (QTPP), which forms the basic design for the development of the product. QTPPs are based on knowledge space developed from the relevant scientific literature with appropriate in vivo relevance; (b) Identification of potential Critical Material Attributes (CMAs), Critical Process Parameters (CPPs), and Critical Quality Attributes (CQAs). As per the ICH Q8 R2 guideline, “CQA is a physical, chemical, biological, or microbiological property or characteristic that should be within an appropriate limit, range, or distribution to ensure the desired product quality”; (c) Perform Risk Assessment (RA) to identify CMAs and CPPs that may have significant impact on CQAs. Among the various tools mentioned in ICH guideline Q9 for performing the RA, Failure Mode Effects Analysis (FMEA) and Failure Mode Effects and Criticality Analysis (FMECA) are the most widely used [34]. These tools help to identify the CMAs and CPPs that affect CQAs. The Design of Experiment (DOE) is based on the RA results. DOE can be planned by considering the most influential CMAs and CPPs defining the design space. As per the regulatory agency United States Food & Drugs Administration (USFDA), changes within the design space is not considered a change [35], and hence it can be used during scale-up and post-approval changes (SUPAC); (d) For ensuring consistent product quality, the development of a control strategy is mandatory, and (e) Finally, product lifecycle management.

In conclusion, applying QbD in the development of a pharmaceutical product helps in better product and process understanding. Therefore, the development of PTH-LPs using the QbD approach was initiated as per guidance ICH Q8(R2) for pharmaceutical development. Further, guidance for the industry for liposomal drug products approved by the Food and Drug Administration (FDA) was also considered during the development of the liposomal formulation [36]. 

Identification of QTPP and CQA (Table 1) was carried out based on prior experience and scientific literature for liposomal formulation development. Figure 1 depicts the overview of the QbD approach used in the development of liposomal formulation, whereas Appendix A summarizes the road map for formulation development of liposomal formulation using the ethanol injection method. 

After the experimentation and data analysis, the obtained experimental results were compared with predicted responses (CQAs/dependent variables) and the percentage residual value was calculated using the following formula;
% Residual = (Predicted results − Observed results)/(Predicted results) × 100(1)

### 2.2. Development of PTH-LPs

PTH-LPs were prepared by the ethanol injection method. Briefly, absolute-ethanol-containing lipid, cholesterol, and DSPE-PEG-2K were injected into 3 mL of acetate-buffer containing PTH (1-34). The resulting mixture was then stirred for a specific period and centrifuged for 60 min at 50,000 rpm. The obtained pellet was redispersed into Milli-Q water and stored at −20 °C. Figure 2B represents the experimental setup for the development of PTH-LPs. Fluorescent PTH-LPs were prepared similar to the aforementioned method, with the addition of NBD-PE lipid (0.5 mM) in organic phase. 

### 2.3. Evaluation of PTH-LPs

#### 2.3.1. Particle Size and Zeta Potential Determination of PTH-LPs

Malvern Zetasizer Nano ZS 90 (Malvern Instrument, Malvern, UK) was used to measure the particle size of optimized PTH-LPs using the photon correlation scattering (DLS) method. Prior to size assessment, the optimized PTH-LPs were diluted with Milli-Q water to achieve the proper scattering intensity. Measurements were made using 1 mL of diluted PTH-LPs in a single-use polystyrene cuvette at 25 °C with a 90° scattering angle. Zeta potential of the PTH-LPs was measured without dilution of samples at 25 °C in triplicate. 

#### 2.3.2. Determination of %EE

The drug content in PTH-LPs was determined using HPLC (HPLC 1260 Infinity, Agilent Technologies, Santa Clara, CA, USA) by injecting the supernatant after ultracentrifugation of PTH-LPs at 50,000 rpm. Chromatographic separation was carried out using the XBridge BEH C18 column (300 Å, 4.6 mm × 150 mm, 10 μm). The mobile phase consisted of 0.1% *v*/*v* formic acid in Milli-Q water as an aqueous phase (A) and 0.1% *v*/*v* formic acid in acetonitrile as an organic phase (B). The injection volume was kept at 50 μL, and analysis was performed at a 210 nm wavelength [40]. %EE was calculated with the help of the following equations.
%EE = (Amount of total drug − Amount of unentrapped drug)/(Amount of total drug added) × 100(2)

#### 2.3.3. Cryogenic Field Emission Scanning Electron Microscopy (Cryo FE-SEM)

Cryo FE-SEM (SIGMA S300, Zeiss, Jena, Germany) was used to inspect the morphology of optimized PTH-LPs. For this, samples were transferred on rivets mounted on a cryo-SEM sample holder. It was then submerged in liquid nitrogen for freezing. The frozen samples were fractured using a cold knife. Sublimation of samples was done for 10 min at −90 °C, and images of samples were obtained after subjecting the samples to the chamber of Cryo FE-SEM.

#### 2.3.4. In Vitro Drug Release

The in vitro PTH (1-34) release from PTH-LPs was performed using a sample-separate method. Pellets of PTH-LPs were dispersed in 10 mL of PBS (pH 7.4). Then, 1 mL of liposomal dispersion was added into single microcentrifuge tubes (MCTs) for a single time point. All MCTs were incubated at 37 °C in an orbital shaker at speed of 100 rpm for 24 h. At time intervals of 5 min, 15 min, 30 min, 1 h, 3 h, 6 h, 12 h, and 24 h, MCTs were taken out of the orbital shaker and centrifuged at 50,000 rpm. The supernatant was removed, and pellets were dissolved in absolute ethanol. Samples were injected in HPLC to quantify released PTH (1-34) from PTH-LPs using a validated HPLC method.

#### 2.3.5. Cell Culture

In present study, MG-63 osteoblast-like cells (NCCS Pune, Pune, India) were used for cellular assay. Cells were cultured in alpha-modified Eagle’s medium (αMEM, Gibco, Billings, MT, USA). The medium was supplemented with 2 mM glutamine, 100 mg mL^−1^ of streptomycin, 100 U mL^−1^ of penicillin, and 10% fetal bovine serum (Invitrogen^TM^, Germany). The cultures were maintained in T-75 flasks and incubated at 37 °C with 5% CO_2_.

##### Cytotoxicity Assay

The cytotoxicity of PTH-LPs was determined by Alamar blue assay. This is a quantitative calorimetric assay based on the biochemical reduction of dye (Alamar blue) from non-fluorescent blue to fluorescent red. Briefly, MG-63 cells were seeded onto 96 well plates at density 1 × 10^4^ cells per well in 100 μL of culture medium. The cells were incubated in an incubator at 37 °C for 24 h for attachment. After 24 h, the culture medium was removed, and cells were washed with PBS. Next, 100 μL of medium containing various concentrations of PTH (1-34) and PTH-LPs were added to a well plate and incubated for 24 h. After 24 h incubation, the medium was removed, and wells were rinsed with PBS. Then, the 100 μL medium containing 10 μL Alamar Blue reagent was added in each well and incubated for another 2 h. Absorbance was measured using a multimode UV microplate reader (Varioskan LUX, Thermo Fischer Scientific, Waltham, MA, USA) at 520–590 nm. The % cell viability was obtained from the following equation:% Viability = (Average of samples treated with formulation)/(Average of control) × 100(3)

##### Cell Proliferation Assay

Cell proliferation assay was performed to evaluate the osteoblast differentiation potential of PTH-LPs on MG-36 cells. For this study, 1 × 10^4^ cells were seeded into 48 well plates. The cells were incubated for 24 h at 37 °C. Then, the cells were treated with PTH-LPs and further incubated for 7 days. At 3, 5, and 7 days, cells were washed with PBS and treated with a 400 μL medium containing Alamar Blue reagent. After 2 h of incubation, absorbance was measured using a multimode UV microplate reader at 520–590 nm. Furthermore, light microscope images of cells treated as a control and samples on the third day of the experiment were also captured to evaluate the osteoblast proliferation potential of PTH-LPs.

##### Calcium Calorimetric Assay

Free calcium concentration from medium after the 7-day incubation period of MG-63 cells was performed using a Calcium Calorimetric Assay Kit (Sigma-Aldrich, Gillingham, UK). Calcium ion concentration was measured by measuring the chromogenic complex formed between o-cresolphthalein and calcium ion. The chromogenic complex was determined spectrophotometrically by measuring the optical density at 575 nm. The optical density was proportional to the concentration of calcium ions present in the samples.

The assay was performed as per manufacturer’s instructions. Briefly, 50 μL of samples or calcium standards were added to a 96-well plate. 90 μL of chromogenic reagent and 60 μL of calcium assay buffer was added to wells containing controls, samples, or standards. The assay plate was incubated for 10 min at room temperature and absorbance was measured at 575 nm using microplate reader.

##### Mineralized Nodule Formation

Alizarin Red assay was used to detect the osteoblast’s mineralized nodule formation upon treatment of the drug and PTH-LPs. For this, cells were seeded at 1 × 10^4^ in 6-well plates and cultured in osteoblast-specific media. After the 24 h, cells were washed with PBS, and different treatments were given. Wells containing only media were considered control. After 7 days, cells were fixed with 4% paraformaldehyde for 15 min. Paraformaldehyde was removed and cells were washed with PBS. Cells were stained with Alizarin Red solution (1.5%) for 15 min. Unreacted dye was removed, and cells were rewashed 2–4 times with PBS. Then, mineral nodules were observed under a light microscope. For quantitative analysis, pyridinium chloride (10%) was added to each well to dissolve the dye, and the concentration of dye was determined by measuring the absorbance of the sample using a microplate reader at 562 nm.

##### Cellular Uptake Study

A cellular uptake study of PTH-LPs was performed by using confocal microscopy. MG-63 cells (1 × 10^5^) were cultured on a glass coverslip in a 35 mm culture plate for 24 h for this study. After 24 h, cells were washed with PBS and incubated with fluorescent PTH-LPs for 1, 3, and 6 h. After each time point, cells were washed with PBS and fixed with 4% (*w*/*v*) paraformaldehyde for 15 min at room temperature. Again, cells were washed 3 times with PBS. The cell nuclei were stained with DAPI for 15 min, and excess DAPI was washed out using PBS three times. The glass coverslip was removed from the culture plate and air dried. DPX mounting agent was used for the attachment of the coverslip on the glass slide and kept at room temperature for drying. Confocal microscope analysis was performed using a laser-scanning confocal microscope (Leica, Wetzlar, Germany). Fluorescent PTH-LPs excitation and emission wavelength were 463 nm and 536 nm respectively. Image J software was used to process the images and to perform semiquantitative analysis. Corrected Total Cell Fluorescence (CTCF) was determined to compare cellular uptake of PTH-LPs at different time points without further processing.

### 2.4. Statistical Analysis

The statistical analysis of the results was carried out using GraphPad Prism (version 6.0, San Diego, CA, USA). All values are expressed as the mean ± standard deviation (SD) (*n* ≥ 3). Statistical significance was determined using Student’s *t*-test. The calculated *p*-values were defined as follows: * *p* < 0.05, ** *p* < 0.01, *** *p* < 0.001. Nonsignificant differences are marked as n.s.

## 3. Results and Discussion

### 3.1. QbD Approach: SETTING of QTPP, Identification of CQAs, and Risk Assessment 

QTPP is the most important element in the QbD approach to help formulation and process design and to understand the final product quality profile. As per ICH Q8 (R2) (2009), QTPP is ‘a prospective summary of the quality characteristics of a drug product that ideally will be achieved to ensure the desired quality, taking into account of safety and efficacy of the drug product’. Table 1 defines the possible QTPP of liposomal products. 

The present study deals with the development of pegylated nanoliposomes encapsulated with the peptide drug PTH (1-34) for systemic administration that would remain in the circulatory system for a prolonged period and deliver the drug for a longer time. Hence, the appropriate size of nanoliposomes (100–200 nm), pegylation of liposomes for longer systemic circulation, the high therapeutic concentration of the drug for therapeutic efficacy, and optimal cholesterol concentration for vesicles stability are the main QTPPs for developing the formulation. After the QTPP, the second most important step of the QbD approach is to determine crucial and potential CQAs of the drug product. CQAs are physical, chemical, biological, and microbiological parameters that should be within the defined range to determine product quality. The selection of CQAs and QTPP is based on an understanding of drug products, preliminary studies, literature review, and experience in formulation development. Risk assessment is a process to determine which CMAs and CPPs affect CQAs.

Risk analysis can be done using several techniques to identify risk factors that influence CQAs. A risk assessment strategy assists in improving the quality of the process and formulation and in identifying the critical attributes that might affect the final formulation’s quality. In this study, we have utilized the Ishikawa fishbone diagram for the identification of risk factors (Appendix A). As per ICH guideline Q9, the risk is a combination of the probability of occurrence of harm and the severity of that harm. Further, a risk estimation matrix was developed to evaluate the qualitative impact of CMAs and CPPs on the CQAs which was identified through the Ishikawa fishbone diagram. In the present work, particle size, %EE, and zeta potential were identified as CQAs of the final product. Thus, understanding the potential risk affecting these CQAs is a crucial step in risk assessment. Table 2 represents the risk estimation matrix showing the risk ranking of CMAs and CPPs based on the Ishikawa fishbone diagram. 

### 3.2. Box Behnken Design 

Based on the preliminary experimentation, prior experience, and risk analysis, lipid concentration, cholesterol concentration, and stirring rate were selected to optimize the PTH-LPs, whereas DSPE-MPEG-200K, stirring time, aqueous to organic ratio, and injection rate were kept constant. For determining the effects of identified CMAs and CPPs, a 3-level Box Behnken Design (BBD) with three independent factors was developed using Design-Expert software (Version 12, Stat-Ease, Inc., Minneapolis, MN, USA). Two different CQAs (responses/dependent factors) were evaluated; particle size and %EE. Levels of independent variables examined in this experimental design were obtained within the range obtained from preliminary experimentation and a literature search. The matrix of experimental design included 17 runs with three replicates. The layout of the experimental design with experimental runs obtained is enlisted in Appendix A. 

Various statistical modules from Design Expert software were utilized to fit the experimental data with the selected experimental design and calculate the statistical parameters. The analysis of variance (ANOVA) method was used for the calculations and data analysis of statistical parameters. Design space establishment is based on the regression models; therefore, the same software was used to generate the design space. All the QTPP criteria are met at a predetermined degree of risk. 

### 3.3. The Influence of CMA and CPP on Particle Size 

The lipidic vesicular size was obtained between 39.48 to 155.83 nm. Statistical analysis showed a model F-value of 5.86, and *p*-values less than 0.0500 indicate the model and its terms are significant. There is only a 1.42% chance that a large F-value could occur due to noise. Values greater than 0.1000 indicate that the model terms are not significant. If there are many insignificant model terms, then model reduction may improve the model. Therefore, a reduced quadratic model was used in particle size analysis to get significant model and model terms. Data obtained from ANOVA showed CMA has a significant impact on particle size, since the *p*-value of the model and lack of fit were 0.0142 and 0.9486, respectively. The lack of fit F-value of 0.29 implies the lack of fit is not significant relative to the pure error. Nonsignificant lack of fit is good. Furthermore, the fit’s statistical data show that the predicted R^2^ of 0.2234 is in reasonable agreement with the adjusted R^2^ of 0.3778, and the difference found between them is less than 0.2. Adequate precision is another statistical parameter that needs to be evaluated that measures the signal-to-noise ratio. A ratio greater than 4 is desirable. In the case of particle size, this ratio is 6.991, indicating an adequate signal,; hence, this model can be used to navigate the design space. Statistical data of model terms are enlisted in Appendix A. 

Furthermore, residual analysis was performed to find the trend of the data sets. This analysis showed that all residual plots behaved very well (Appendix A). Residuals estimate the experimental errors obtained by subtracting the observed responses from the predicted responses. In this formulation, lipid concentration had a significant influence/impact on particle size; this affirmation was sustained by counter plots and the 3D surface plot (Figure 3 and Appendix A). This demonstrates that a rise in lipid concentration increases the vesicular size. Cholesterol concentration showed the curvature effect. When cholesterol concentration is at 10–12 mM, it shows an increase in the particle size. Furthermore, a concentration between 13–17 mM of cholesterol showed an unchanged particle size. Generally, an increase in the stirring speed reduces the particle size, but in the present work, we did not find an impact of stirring speed; this might be due to the spontaneous formation of liposomes in the ethanol injection method or the narrow range of the stirring speed. The box plot depicted in Appendix A is indicative of the possible interaction of all the variables on the defined response.

### 3.4. The Influence of CMA and CPP on %EE

To prepare nanovesicles, lipid and cholesterol solution was injected into the drug-containing aqueous solution. Therefore, the lipid solution’s molar concentration significantly impacts the final liposomal drug concentration and %EE. %EE was found to be between 24.58% and 87.98%. Statistical analysis showed the model F-value of 4.88, and *p*-values of 0.0432 indicate model and models’ terms are significant. There is only a 4.32% chance that an F-value this large could occur due to noise. For %EE, a reduced linear model was used to obtain the significant model and model terms. The lack of fit F-value of 0.59 implies the lack of fit is not significant relative to the pure error. There is a 77.61% chance that a lack of fit F-value this large could occur due to noise. The difference between predicted R^2^ and adjusted R^2^ is less than 0.2. A ratio of 4.552 shows an adequate signal, suggesting this model is adequate for navigation of the design space. Appendix A enlists all the statistical terms for model selection. Similar to particle size, the residual analysis was also carried out for %EE. All the residual plots behaved well. Appendix A depicts all residual plots for %EE. 

An increase in lipid concentration may increase the %EE. In the current formulation, lipid concentration had a significant role in %EE. Increased lipid concentration showed an improvement in %EE, which was depicted from all counter plots and 3D surface plots (Figure 3 and Appendix A). It has been observed that high cholesterol content may negatively influence %EE, and thus the optimum concentration of cholesterol could be obtained. In our case, with an increase in the cholesterol concentration, a reduction in %EE was observed, which might be due to the competition of hydrophobic cholesterol with hydrophilic peptide drug. Appendix A depicts the box plot for %EE. 

### 3.5. Design Space Generation

Considering the target range of CQAs, a design space was generated (Appendix A) by considering the impact of formulation variables, i.e., lipid concentration and cholesterol concentration. The stirring speed was set constant at 500 rpm to finalize the formulation strategy.

For the optimal formulation, the targeted CQAs were maximum entrapment of peptide into the lipid vesicles and a particle size less than 200 nm. The optimal formulation was obtained by using 60.25 ± 11.64 mM of lipid, 9.33 ± 2.57 mM of cholesterol, 3 mM of pegylated lipid, and 500 rpm stirring speed. Table 3 represents the optimized parameters for the developed liposomal product. The optimal formulation was developed in triplicate. The obtained experimental CQAs, expressed as mean values, were compared with predicted values (Table 3). The experimental results were found close to the predicted values. Furthermore, the predictability of the model was demonstrated by % residual values.

### 3.6. Physicochemical Evaluation of PTH-LPs

#### 3.6.1. Particle Size, Zeta Potential, and Morphological Assessment

Physicochemical characteristics such as particle size, zeta potential, and entrapment efficiency of PTH-LPs were evaluated. The particle size distribution of the optimized PTH-LPs is presented in Figure 4A. The single peak of PSD of PTH-LPs demonstrated the even distribution of liposomal vesicles in the optimized PTH-LPs. Average particle size of optimized PTH-LPs was found to be 147.76 ± 2.14 nm, and zeta potential of PTH-LPs was found to be −27.00 ± 3.55 mV (Figure 4B). %EE of PTH-LPs exhibited 69.18 ± 3.62%. The developed PTH-LPs were also characterized by Cryo FE-SEM. This visual technique was employed to obtain details related to surface morphology and the mean size of liposomes. The images revealed the spherical shape of liposomes having a smooth surface. Figure 4C,D represents the morphological assessment of optimized PTH-LPs.

#### 3.6.2. In Vitro Drug Release

In vitro drug release of plain PTH (1-34) and PTH (1-34) from PTH-LPs is shown in Figure 5. PTH-LPs demonstrated the sustained release of PTH (1-34) in PBS (pH 7.4), whereas drug release from plain PTH (1-34) solution showed more than 95% of the drug was released within 5 min. From PTH-LPs, more than 50% of the drug was released from formulation at 60 min, which was increased up to 70% at 3 h. Plateau release profile was observed between 3 h to 6 h. Furthermore, it reached 75% at 6 h. More than 85% of the drug was released after 24 h. This sustained drug release pattern of PTH (1-34) from PTH-LPs might be due to the lipidic barrier of liposomes hindering the diffusion of PTH (1-34). Moreover, cholesterol used in the formulation improves lipid bilayer fluidity and prevents the further diffusion of the drug [41].

#### 3.6.3. In Vitro Cytotoxicity Study and Cell Proliferation Assay

The in vitro cytotoxicity of free drug and PTH-LPs towards MG-63 cells was evaluated using Alamar Blue assay. As shown in Figure 6, % viability of the cells treated with the drug and PTH-LPs for 24 h was evaluated at a concentration up to 40 µg/mL. Similar to the PTH (1-34) (Figure 6A), PTH-LPs showed less cytotoxicity to the cells. Hence, these are safe as a biocompatible carrier for drug delivery. These results confirm that the PTH-LPs are safe in anti-osteoporotic therapy.

Furthermore, as a part of the cell proliferation assay, cells were incubated for 7 days after the treatment. At 3-, 5-, and 7-day intervals, % proliferation was calculated using Alamar Blue reagent (Figure 7A–C). The cells treated with PTH-LPs showed a visibly high number of cells compared to the control and drug alone. These results could be due to the anabolic effect of sustained peptide release from PTH-LPs. Moreover, the osteoblast secretes the organic matrix, and in later stages, this matrix is mineralized by various biochemical pathways [42]. Therefore, this matrix could be the reason for the entrapment of the PTH-LPs and the formation of a depot. This depot probably releases the drug for a prolonged period of time; hence, there was more proliferation in the PTH-LPs group than in the other groups. Figure 7A demonstrates the increase in MG-63 cell numbers in terms of % proliferation, while Figure 7B depicts the photographic images of 3-day incubation of MG-63 cells. Figure 7C represents the line graph for % cell proliferation. These results suggest that PTH-LPs release the anabolic peptide in an anabolic dose, which is evident from the proliferation of MG-63 cells.

#### 3.6.4. Calcium Calorimetric Assay

Calcium calorimetric assay (Figure 8) shows the efflux or influx of calcium ions due to the anabolic or catabolic activity of the PTH (1-34) peptide. Based on the dose, PTH stimulates the influx or efflux of calcium. The cells treated with PTH-LPs had more calcium efflux, whereas the cells treated with bare peptide and the control group showed less compared to the PTH-LPs. These results along with the cell proliferation data demonstrate the anabolic effect of peptide on osteoblast cells.

#### 3.6.5. Mineralized Nodule Formation

Osteoblast mineralization can be assessed by quantitative mineralization experiments such as Alizarin Red assay. Therefore, Alizarin Red assay was performed to determine the mineralization by MG-63 cells after the incubation with plain drug and PTH-LPs. Cells without treatment were considered the control. 1.5% Alizarin Red was used for staining purposes. Figure 9A demonstrates the number of red mineralization nodules (red frame) in the three experimental groups. The PTH-LPs showed significantly higher numbers of nodules than PTH (1-34) and control. Figure 9B depicts the microscopic images of the mineralized nodules that developed after the treatment of different experimental groups. For quantitative analysis, nodules were dissolved at 10% pyridinium chloride, and absorbance was measured (Figure 9C). As seen in the proliferation assay, % cell proliferation was observed more in PTH-LPs compared to the control and drug alone. These higher numbers of cells might have released the excess calcium which was observed in the calcium calorimetric assay. Improved proliferation and calcium ion concentration could cause the improved mineralization of MG-63 cells after the treatment.

#### 3.6.6. Cellular Uptake Study

The cellular uptake of PTH-LPs was evaluated semi quantitatively by fluorescence microscopy. Figure 10 demonstrates the cellular uptake of PTH-LPs. For this, fluorescent PTH-LPs were incubated with MG-63 cells for different time intervals. Cell nuclei were stained with 4′,6-diamidino-2-phenylindole (DAPI). Fluorescence signal of PTH-LPs delivered to the cells was observed in cell cytoplasm at 1, 3, and 6 h. As the time of incubation increased from 1 h to 6 h, the cell fluorescence grew brighter. At 6 h, significant fluorescence was observed compared to 1 and 3 h (Figure 10A). Figure 10B depicts the enlarged images after 6 h of treatment. Furthermore, semiquantitative fluorescence was measured by Image J software (Figure 10C). As shown in Figure 10C, cellular uptake was observed to be closely related to the length of incubation of the cells.

## 4. Conclusions

The objective of the present study was to formulate PTH-LPs that could overcome the stability-related issues of the anabolic peptide. Based on the data obtained from QbD, it has been shown that this approach is the key element in the development of PTH-LPs by providing information on the CMAs and CPPs during formulation development. This approach showed that lipid concentration has a significant influence on %EE and particle size. Optimized formulation was achieved by formulating the suggested composition from software, whose quality attributes are within the specified limits. PTH-LPs showed a sustained drug release profile up to 24 h. In vitro cell experiments revealed that PTH-LPs had excellent biocompatibility with MG-63 cells. Cell proliferation and mineralized nodule formation assay suggested that PTH-LPs induced osteoblast differentiation and mineralization as well as the formation of mineralized nodules. Furthermore, the cell uptake study revealed that the PTH-LPs have a great capacity for cellular uptake in MG-63 cells. Hence, it is concluded that PTH-LPs could be a potential solution for effectively delivering PTH (1-34) in osteoporosis.

## Figures and Tables

**Figure 1 pharmaceutics-14-02417-f001:**
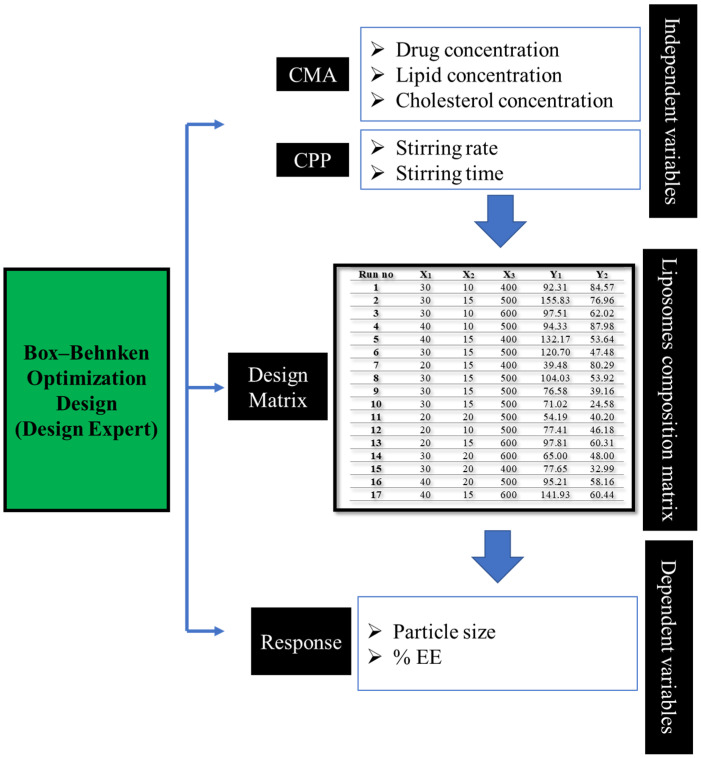
Overview of QbD approach for development of PTH-LPs, where X_1_–X_3_ and Y_1_–Y_2_ are factors and responses respectively.

**Figure 2 pharmaceutics-14-02417-f002:**
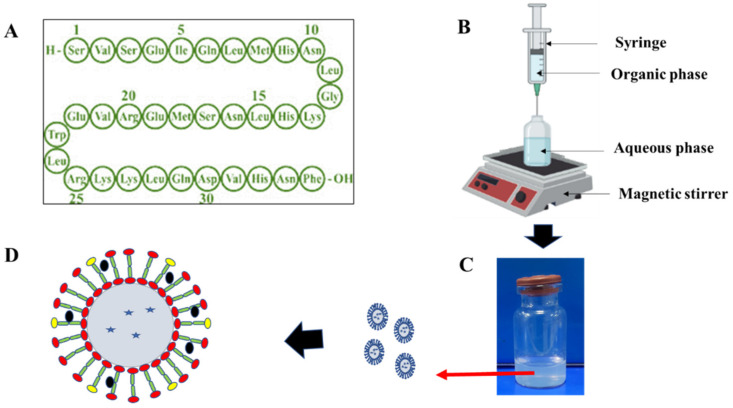
Development of PTH-LPs. (**A**) Amino acid sequence of PTH (1-34), (**B**) Representative instrumental setup for the development of PTH-LPs, (**C**) Vial containing dispersion of PTH-LPs, (**D**) Schematic representation of PTH-LPs.

**Figure 3 pharmaceutics-14-02417-f003:**
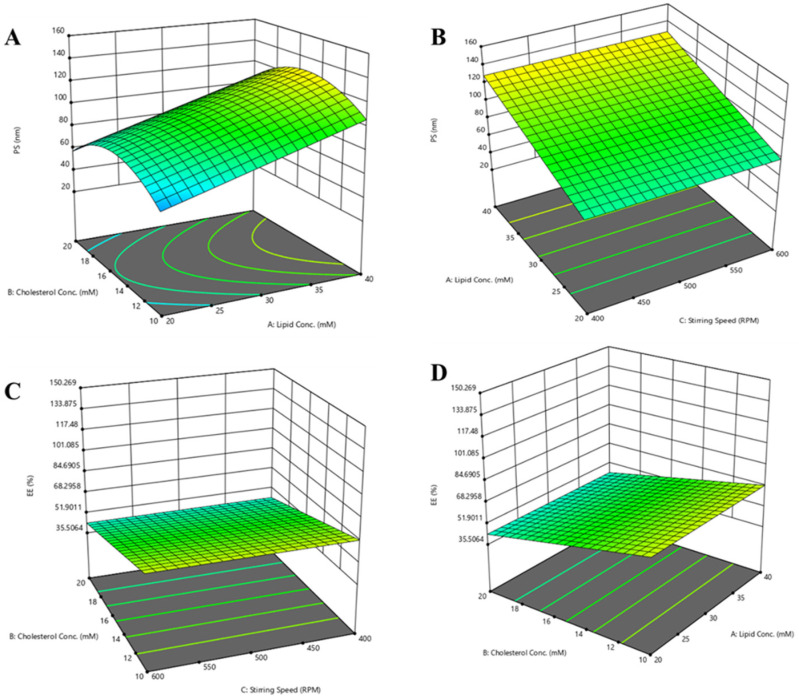
Three-dimensional surface plot for PTH-LPs. (**A**) Effect of lipid concentration and cholesterol concentration on particle size. (**B**) Effect of stirring speed and lipid concentration on particle size. (**C**) Effect of lipid concentration and stirring speed on %EE. (**D**) Effect of lipid concentration and cholesterol concentration on %EE.

**Figure 4 pharmaceutics-14-02417-f004:**
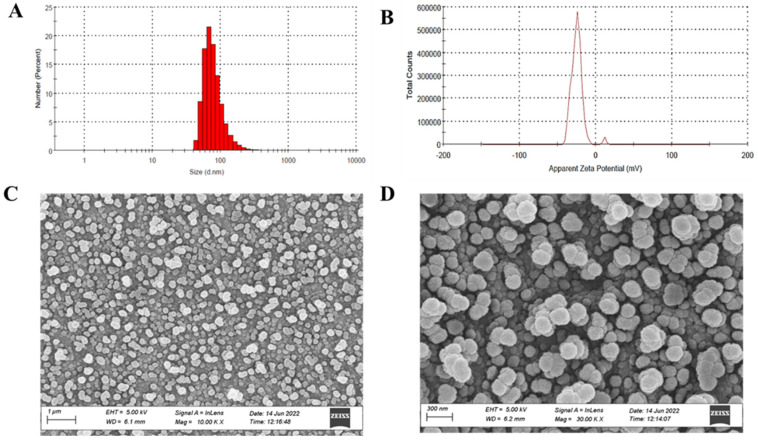
Characteristics of optimized PTH-LPs. (**A**) Particle size distribution, (**B**) Zeta potential, (**C**,**D**) Morphological assessment by Cryo-SEM. The data represent the means ± SD (*n* = 3).

**Figure 5 pharmaceutics-14-02417-f005:**
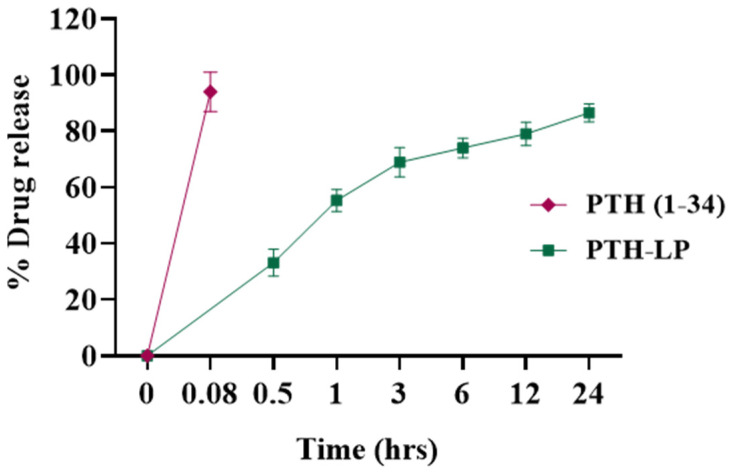
In vitro release profile of PTH (1-34) (red) and PTH (1-34) from PTH-LPs (green). Data represent the mean ± SD (*n* = 3).

**Figure 6 pharmaceutics-14-02417-f006:**
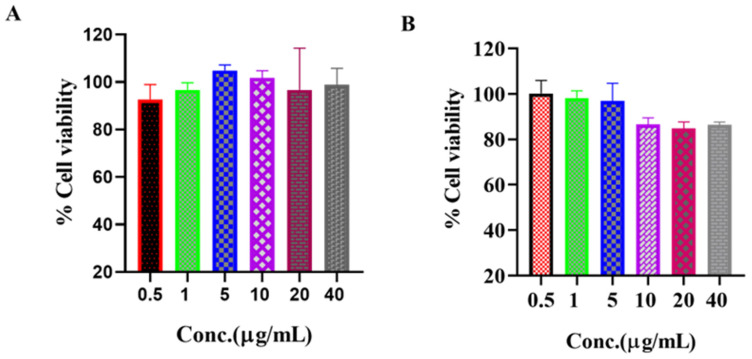
Cytotoxicity study on MG-63 cells. (**A**) % cell viability of PTH (1-34) and (**B**) % cell viability of PTH-LPs at different concentrations.

**Figure 7 pharmaceutics-14-02417-f007:**
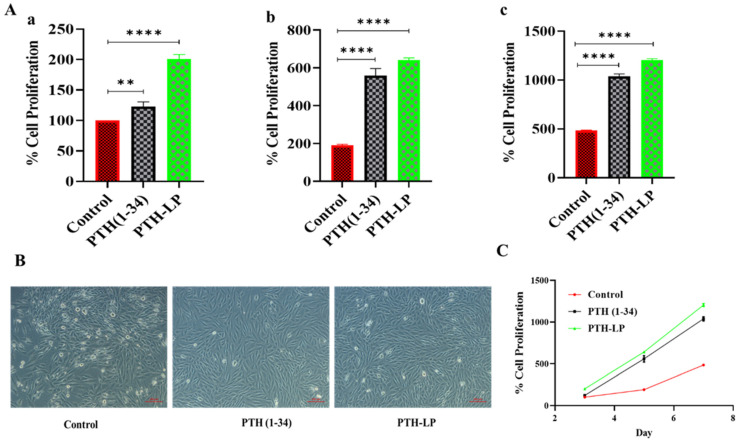
Cell proliferation assay of PTH-LPs. (**A**) % cell proliferation of PTH-LPs on MG-63 cells at 3 (**Aa**), 5 (**Ab**), and 7 (**Ac**) days. (**B**) Microscopic images of MG-63 cell after 3 days of incubation. (**C**) Line graph for % cell proliferation of MG-63 cells after 3, 5, and 7 days of incubation. Scale bar: 100 μm. ** *p* < 0.01, **** *p* < 0.0001.

**Figure 8 pharmaceutics-14-02417-f008:**
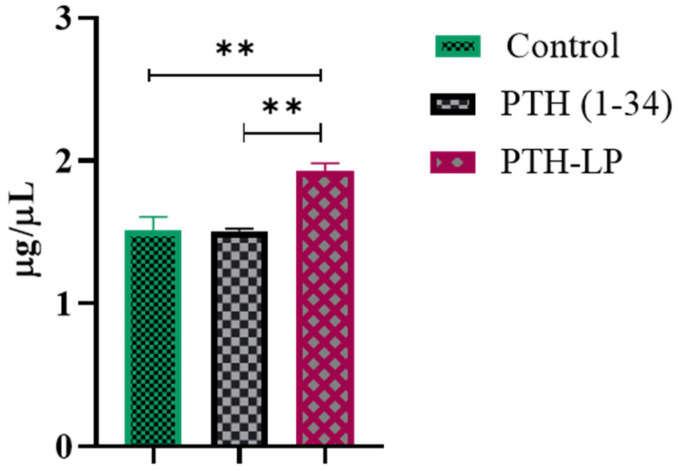
Calcium calorimetric assay. The results represent the means ±SDs (*n* = 6). ** *p* < 0.01.

**Figure 9 pharmaceutics-14-02417-f009:**
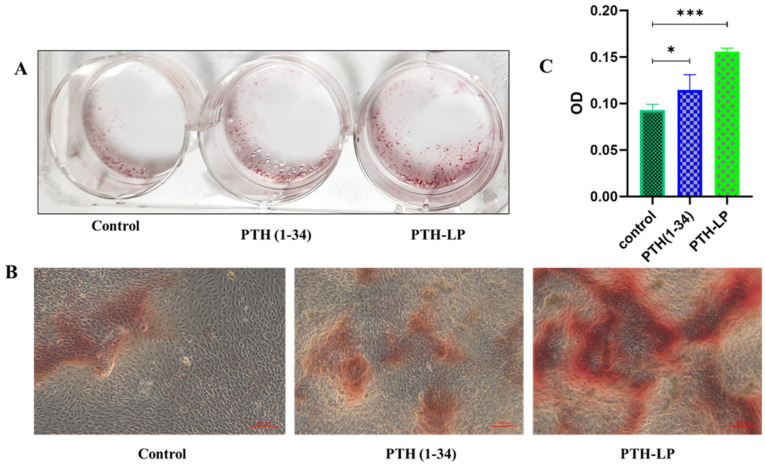
Assessment of mineralized nodule formation after the treatment of experimental groups. (**A**) Photographs of mineralized nodules (scale bar 100 μm), (**B**) Microscopic images of mineralized nodules, and (**C**) Quantitative measurement of developed mineralized nodules. * *p* < 0.05, *** *p* < 0.001.

**Figure 10 pharmaceutics-14-02417-f010:**
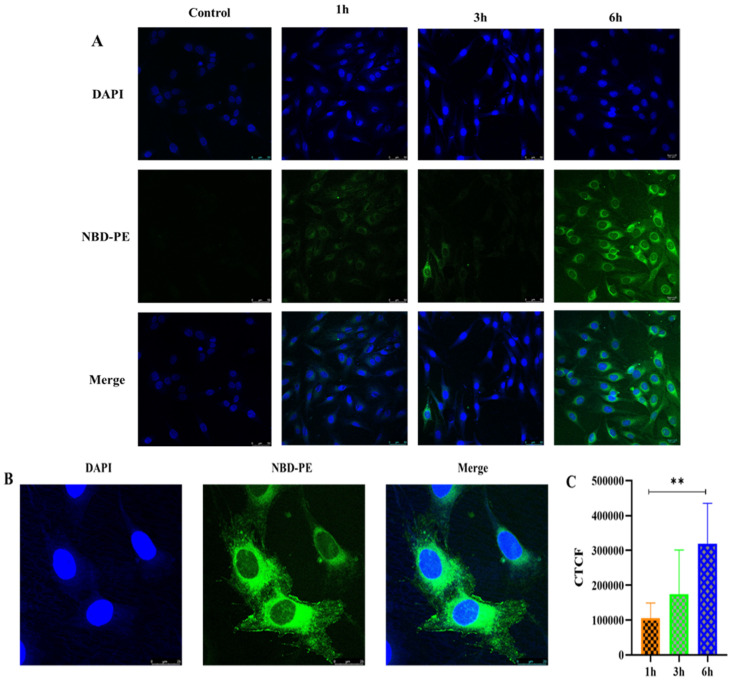
Cellular uptake in MG-63 cells incubated with fluorescent PTH-LPs. (**A**) Confocal microscopic images for time-dependent cellular uptake study (scale bar 50 μm), (**B**) Cellular uptake at 6 h (scale bar 25 μm), (**C**) Corrected Total Cell Fluorescence (CTCF) after treatment of PTH-LPs at 1 h, 3 h, and 6 h. ** *p* < 0.01.

**Table 1 pharmaceutics-14-02417-t001:** QTPPs and CQAs for PTH-LPs.

QTPPs				
QTPP Element		Target	Justification	Ref
Dosage form		Ready to use	Patient convenience and cost consideration	
Dosage design		Liposomes	Scale-up feasibility as well as high safety of excipients used in manufacturing of liposomes	
Drug product quality attributes	Drug content (%EE)	>30%	A maximum %EE is associated with minimum drug loss during the manufacturing process which ultimately reduces the production cost of formulation	[37]
	Size	<200 nm	A smaller vesicle size is desired to escape from the RES system upon systemic administration of the formulation	[38]
	>−30 mV/+30 mV	>−30 mV/+30 mV	Higher or lesser values impart the repulsion of vesicles from each other and hence improve the stability of the formulation	[39]

**Table 2 pharmaceutics-14-02417-t002:** Risk estimation matrix.

	Parameters	Drug Product CQA’s		
		%EE	Particle Size	Zeta Potential
**CMAs**	Drug concentration	High	Medium	Medium
	Lipid concentration	High	High	Medium
	Cholesterol concentration	High	High	Medium
**CPPs**	Stirring rate	High	High	Medium
	Stirring time	Medium	Medium	Low

**Table 3 pharmaceutics-14-02417-t003:** Optimized formulation parameters.

CMAs/CPPs	Values	Responses	Target	Predicted Value	Experimental Value	Residual Values (%)
Lipid concentration (mM)	60.25 ± 11.64	Y_1_	<200 nm	142.93 ± 2.49	147.76 ± 2.14	−3.26
Cholesterol concentration (mM)	9.33 ± 2.57	Y_2_	>30%	70.63 ± 6.51	69.18 ± 3.62	2.05

## Data Availability

Not applicable.

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
