# Peer review of "Anabolic Peptide-Enriched Stealth Nanoliposomes for Effective Anti-Osteoporotic Therapy"

_pharmaceutics, 2022, doi:10.3390/pharmaceutics14112417_

Round 1
Reviewer 1 Report
The manuscript by Sagar Salave, et al. presents a well-organized study in the preparation of a PTH-liposome nanosystem. Several parameters have considered and optimized during the preparation. In addition, a primary biological test has been conducted. This research can indeed offer useful information for the readers who are working in the related fields. I would like to see it will be published in Pharmaceutics after addressing some issues.
1. High-speed centrifugation can sometimes lead to instability to the liposomal structures. Also, full redispersion is often difficult as those liposomal pellets stick together. The authors have to mention and discuss if there are such issues during those processes.
2. The DLS data is from the redispersion solution or the original synthesis solution?
3. Stability study is insufficient, the long-term (at least cover the longest testing time in this paper) stability of this liposome solution, those in the cell culture medium and in serum have to be examined.
4. Scale bars in images are not clear, please revise them. The words of scales in figure 9 are too small.
5. The most recently reported liposome-based protein and peptides delivery systems should be cited, such as Chin. Chem. Lett. 2021, 33:587-596; Pharmaceutics, 2020 20;12(10):993.
Author Response
Reviewer 1
- High-speed centrifugation can sometimes lead to instability of the liposomal structures. Also, full redispersion is often difficult as those liposomal pellets stick together. The authors have to mention and discuss if there are such issues during those processes.
Ans: We agree with this point. High-speed centrifugation can sometimes lead to instability of the liposomal structures and full redispersion is often difficult as those liposomal pellets stick together. However, in our case, we didn’t face these types of instability issues. During the formulation optimization, we carefully optimized the speed and time of the centrifugation. Probably, hydrophilic coating on liposomes due to pegylated lipids in formulation could have prevented the frequent aggregation seen in non-pegylated liposomes and helped here to maintain the stability of liposomes [1].
Ref. [1] J.S. Suk, Q. Xu, N. Kim, J. Hanes, L.M. Ensign, PEGylation as a strategy for improving nanoparticle-based drug and gene delivery, Advanced Drug Delivery Reviews. 99 (2016) 28. https://doi.org/10.1016/J.ADDR.2015.09.012.
- The DLS data is from the redispersion solution or the original synthesis solution?
Ans: DLS data provided in the manuscript is from the redispersion solution. We also wish to bring to notice of reviewer that we have data of original dispesion which are not significantly different fom the redispersion data,again suggesting that centrifugation process does not have any detrimental effect on the stability of developed liposomal compositions.
- The stability study is insufficient, the long-term (at least cover the longest testing time in this paper) stability of this liposome solution, those in the cell culture medium and in serum have to be examined.
Ans: We appreciate the reviewer’s this keen observation. We have used the fetal bovine serum for cell line studies. As per our best understanding, the serum contains 60–80 mg/mL protein content. Therefore, such samples are not recommneded to be injected into the HPLC as such and required extensive sample preparation techniques before it can be injected. Furthermore, the drug concentration used is very less in the culture medium and hence posing a challenge in quantify the same using HPLC. Moreover, we would like to bring to the reviewer’s notice that we are optimizing the lyophilization for developed formulation and we have planned an extensive stability studies which can be included in our future research article which would be based on the in vivo pharmacokinetic and toxicological studies for optimized liposomes.
- Scale bars in images are not clear, please revise them. The words of scales in figure 9 are too small.
Ans: As per the reviewer’s comment, images have been enlarged in the revised manuscript. Further, scale bar also included in figure legend.
- The most recently reported liposome-based protein and peptide delivery systems should be cited, such as Chin. Chem. Lett. 2021, 33:587-596; Pharmaceutics, 2020 20;12(10):993.
Ans: As per the reviewer’s suggestion, suggested references have been included in the revised manuscript and highlighted in yellow color.
Reviewer 2 Report
The paper presents a work of developing method of PTH-LPs by using the Quality by Design approach. The performance of this method was tested in vitro on MG-63 osteoblast-like cells using cytotoxicity, proliferation, calcium calorimetric and mineralised nodule formation and cellular uptake assays. The outcome of this study demonstrated the potential of QbD concept to improve the osteoanabolic therapy in osteoporosis. The method and investigation results are very straightforward. The work is right in the scope and well fitted to be published on the journal of Pharmaceutics. Even though this work is just a prototype of testing on cell-lines, it is still meaningful to the Pharmaceutic community. The reviewer recommend it to be directly accepted for publication with a few careful check of details.
Author Response
Reviewer 2
The paper presents a work of developing a method of PTH-LPs by using the Quality by Design approach. The performance of this method was tested in vitro on MG-63 osteoblast-like cells using cytotoxicity, proliferation, calcium calorimetric, and mineralized nodule formation and cellular uptake assays. The outcome of this study demonstrated the potential of the QbD concept to improve osteoanabolic therapy in osteoporosis. The method and investigation results are very straightforward. The work is right in scope and well-fitted to be published in the journal of Pharmaceutics. Even though this work is just a prototype of testing on cell lines, it is still meaningful to the Pharmaceutic community. The reviewer recommends it to be directly accepted for publication with a few careful checks of details.
Ans: We are thankful to reviewer for the motivational statements.
Reviewer 3 Report
Congratulations for your interesting work!
My comments and question:
1. Please remove the extra space in the following lines: 184,195, 329, 411, 459, 471
2. It would be better to see the figure text right below the Figure 1, not on the next page
3. Please explain why the cell viability significantly lower in case of PTH-LPs from 10-40 ug/ml concentration range than 0,5-5 ug/ml concentration range (Figure 6 B)?
Author Response
Reviewer 3
- Please remove the extra space in the following lines: 184,195, 329, 411, 459, 471.
Ans: As per the reviewer’s suggestion, extra space have been removed in revised manuscript. However, in some splaces extra space was remained due to formate of journals template.
- It would be better to see the figure text right below the Figure 1, not on the next page.
Ans: As per the reviewer’s suggestion, required changes have been done in revised manuscript and highlighted in yellow color. Further, we will make sure that figures caption will be under the figure only during proof stage too.
- Please explain why the cell viability significantly lower in case of PTH-LPs from 10-40 ug/ml concentration range than 0,5-5 ug/ml concentration range (Figure 6 B)?
Ans: We appreciate the reviewer’s this keen observation. We found that high concentration of drug might have toxic effect on MG-63 cells and therefore reduction in % cell viability was observed. However, % cell viability is still under accepted limit as per ISO 10993-5.
Pursuant to ISO 10993-5, percentages of cell viability above 80% are considered as non-cytotoxic; within 80%–60% weak; 60%–40% moderate and below 40% strong cytotoxicity respectively [1].
Ref. [1]. ISO 10993-5:2009 Biological Evaluation of Medical Devices. Part 5: Tests for In Vitro Cytotoxicity. International Organization for Standardization; Geneva, Switzerland: 2009.
We appreciate the suggestions of the reviewers to our original submission and thank you for allowing us to revise the same.